# OpenReview forum: "Certifying LLM Safety against Adversarial Prompting"
_colmweb.org/COLM/2024/Conference — COLM_

### Official Review · Reviewer_bDKt · 2024-05-08

**Rating:** 7
**Confidence:** 3
**Ethics Flag:** 1

**Summary:**

The main contribution of this paper is an erase-and-check framework for certifiable safety against adversarial prompts, coupled with three practical approaches that are tested empirically. In this case, an adversarial prompt is one that has additional tokens that may trick the LLM to responding (e.g., asking for directions to commit fraud; by adding new tokens the LLM may incorrectly respond with directions). The main idea of the overall approach is to test the safety of a prompt and versions of the same prompt with some tokens deleted: if all safety tests are okay, then the prompt is okay. Otherwise, some version is un-safe and so the prompt should be ignored. The paper is in the "certifiable safety" style of work where a formal proof is presented with certain guarantees. The three practical versions of the approach include a random sampling one (where tokens are randomly selected for deletion, then check the prompt), a greedy version, and a gradient-based one. Results show strong performance, even as the adversarial sequence length grows.

Overall, this is a strong paper that makes a nice contribution both to certified robustness and through practical methods that demonstrate strong performance in defending against adversarial prompts. The paper is well-structured -- with the questions percolating in my reading of the paper naturally answered.

**Reasons To Accept:**

+ Timely work on adversarial prompts of LLMs.

+ The paper pairs an intuitive method (erase-and-check) with solid formal work and good empirical studies of its practical effectiveness.

+ The three methods -- random, greedy, and gradient-based -- are evaluated for increasing adversarial sequence lengths with respect to accuracy and running time, giving a good understanding of the trade-offs as well as relative strengths.

**Reasons To Reject:**

- A minor concern is that all the empirical work is based on Llama 2 and DistilBERT. A comparison with others could strengthen the work, but I don't see this as a serious drawback considering the rest of the contributions in the paper.

- Another minor concern is in a deeper analysis of the failure cases for the different methods. I wonder if the gradient approach versus greedy display any patterns in the kinds of adversarial prompts they can deal with. Again, this is a fairly minor point.

---

> ### Author Rebuttal · Authors · 2024-05-31
>
> 1. We evaluate the performance of erase-and-check with GPT-3.5, Llama-2 13B, and Llama-3 8B (the latest Llama model) as the safety filter. We report the certified accuracy on the harmful prompts and the empirical accuracy on the safe prompts in the suffix mode. In addition, we include the corresponding results for the Llama-2 7B and DistilBERT models from our paper for comparison.
>
> | LLM | Cert Acc |
> |------|------------|
> | GPT-3.5 | 100 |
> | Llama-3 8B | 98 |
> | Llama-2 13B | 99 |
> | Llama-2 7B | 92 |
> | DistilBERT | 99 |
>
> Empirical accuracy on safe prompts:
>
> | Max Erase Length | 0 | 10 | 20 | 30 |
> |----------|--|--|--|--|
> | GPT-3.5 | 99 | 87 | 86 | 91 |
> | Llama-3 8B | 100 | 99 | 98 | 98 |
> | Llama-2 13B | 99 | 96 | 96 | 96 |
> | Llama-2 7B | 100 | 98 | 97 | 97 |
> | DistilBERT | 100 | 100 | 100 | 100 |
>
>    The new models we tested demonstrate high certified accuracy (greater than 98%) on harmful prompts. On safe prompts, Llama-2 13B and Llama-3 8B exhibit strong empirical accuracies, comparable to those of Llama-2 7B as reported in our previous paper. Although the empirical performance of GPT-3.5 is lower than that of the Llama models, it remains above 85%. Upon inspecting the outputs of GPT-3.5, we found that this model misclassifies the prompts at a higher rate than the Llama models. It is also important to note that the system prompt used for the safety filter is specifically tuned for Llama models, which may partially explain the lower performance of GPT-3.5. Adjusting the system prompt for GPT-3.5 could potentially improve its performance.
>
>    We will include the above results in the revised version of our paper.
>
> 2. In our analysis, we do not observe any noticeable difference in the type of adversarial sequences the two approaches can deal with. However, upon inspecting the tokens removed by the greedy and gradient-based methods, we observe distinct behaviors. The greedy method predominantly eliminates tokens from the adversarial sequence, allowing it to nearly achieve the certified performance of 99\% after a sufficiently large number of iterations (Figure 7). In contrast, the gradient-based method erroneously removes tokens from the original harmful prompt. Removal of harmful tokens from the original prompt can inadvertently render it safe. Consequently, the gradient-based method remains below the certified performance, even after numerous iterations (Figure 8).

---

> > ### Comment · Reviewer_bDKt · 2024-05-31
> >
> > Thank you for the updates.

---

### Official Review · Reviewer_6h1c · 2024-05-10

**Rating:** 5
**Confidence:** 3
**Ethics Flag:** 1

**Summary:**

This paper proposes a method called erase-and-check to defend against adversarial prompts. The main idea is to test all possible substrings with a safety filter, and the original prompt will be labeled as safe only if none of the subsequences are detected as harmful. To further speed up the detection, three empirical defense methods including RandEC, GreedyEC, and GradEC are proposed. Experimental results are conducted to evaluate the effectiveness.

**Questions To Authors:**

Minors:
certifed defenses --> certified defenses

**Reasons To Accept:**

- Defending against jailbreak prompts is an important research task, which is under studied in existing literature
- The proposed method is intuitive and simple to implement - only requiring a safety filter
- three empircal methods can speed up the detection process and with a reasonable accuracy

**Reasons To Reject:**

- Regarding the novelty: exhausting all possible substrings is hardly can be limited in novelty
- the comparison with past studies is insufficient, most of the experiments are focused on the proposed method and its variants, more experiments with previously method or straightforward methods should be conducted
- The proposed method can only be used for suffix type jailbreak method, which somehow limits its generality - but this is just a minor point

---

> ### Author Rebuttal · Authors · 2024-05-31
>
> 1. The main novelty of our work lies in the introduction of a framework that enables us to design certified defenses against three adversarial attack modes—suffix, insertion, and infusion—and three empirical variants, RandEC, GreedyEC, and GradEC, which allow end users to trade off attack performance with computational efficiency. Our certifiable erase-and-check procedures provide theoretical guarantees in the form of safety certificates and involve a comprehensive search over all subsequences. In contrast, the other approaches mitigate computational complexity through methods such as random subsampling (RandEC), greedy token erasure (GreedyEC), and gradient-informed token erasure (GradEC).
>
> 2. In Appendix F, we compare our method with randomized smoothing (Figure 9).
>
>     We perform additional analyses to compare our method with existing empirical defenses such as perplexity filtering, paraphrasing, and adversarial training [1, 2]. While these defenses demonstrate effectiveness against the GCG attack, they have been shown to fail against newer adversarial attacks, such as AutoDAN-HGA [3]. In contrast, the performance of our erase-and-check is guaranteed to remain unchanged against new attacks, owing to its safety certificate.
>
> | Method | Detection Accuracy (%) |
> |-------|-------|
> | Perplexity Defense (Vicuna) | 2.3 |
> | Perplexity Defense (Guanaco) | 1.5 |
> | Perplexity Defense (Llama 2) | 39.2 |
> | Paraphrasing Defense (Vicuna) | 32.0 |
> | Adversarial Training (Vicuna) | 7.0 |
> | Erase-and-Check (Llama 2) | **92.0** |
> | Erase-and-Check (DistilBERT) | **99.0** |
>
> We will include the above analysis in the revised version of our paper.
>
> [1] Detecting language model attacks with perplexity, Alon and Kamfonas, 2023.
>
> [2] Baseline defenses for adversarial attacks against aligned language models, Jain et al, 2023
>
> [3] Autodan: Generating stealthy jailbreak prompts on aligned large language models, Liu et al, 2023.
>
> [4] Universal and transferable adversarial attacks on aligned language models, Zou et al., 2023
>
> 3. Our method is not restricted to suffix-type jailbreaking attacks and can also handle insertion and infusion attacks (as discussed in Sections 3.2 and 3.3). We describe the erase-and-check procedure for the insertion and infusion modes in paragraphs 4 and 5, respectively, on page 6. Figures 3 and 5 show the performance and running time of erase-and-check for the suffix, insertion, and infusion modes.

---

> > ### Author Response · Authors · 2024-06-06
> >
> > Dear reviewer,
> >
> > We thank you again for taking the time to review our work. As the discussion period ends soon, we seek your feedback on whether we have adequately addressed your concerns in our rebuttal. We addressed concerns regarding the novelty and generality of our method. We also included an analysis comparing the performance of baseline empirical defenses and our certified defense against the AutoDAN-HGA attack, demonstrating that the detection accuracy of the baseline defenses is lower than our certified detection accuracy.
> >
> > Below, we present additional experimental results comparing the performance of our erase-and-check procedure and its empirical variants with baseline defenses against AutoDAN-HGA. We run erase-and-check with Llama-2 and DistilBERT as the safety filter, setting the maximum erase length to 10 tokens, which is significantly shorter than the adversarial sequence length produced by AutoDAN-HGA (going up to 80 ~ 100 tokens). Our observations indicate that the detection accuracy of our method surpasses even the certified accuracy reported in previous results. Furthermore, we evaluated RandEC (sampling ratio = 0.3 and max erase = 30 tokens) and GreedyEC (#iterations = 6) against AutoDAN-HGA and found that their detection accuracy is significantly higher than the baseline defenses. These findings further demonstrate the resilience of our method against novel and unseen adversarial attacks.
> >
> >
> > | Method | Detection Accuracy (%) |
> > | --------- | --------------------------- |
> > | Perplexity Defense (Vicuna) | 2.3 |
> > | Perplexity Defense (Guanaco) | 1.5 |
> > | Perplexity Defense (Llama 2) | 39.2 |
> > | Paraphrasing Defense (Vicuna) | 32.0 |
> > | Adversarial Training (Vicuna) | 7.0 |
> > | **Erase-and-Check (Llama 2)** | **99.0** |
> > | **Erase-and-Check (DistilBERT)** | **100.0** |
> > | **RandEC** | **98.5** |
> > | **GreedyEC** | **100.0** |
> >
> > We will include the above results in the revised version of our paper.
> >
> > Should you have any further comments or questions, please do not hesitate to let us know. If all concerns have been addressed, we would greatly appreciate your consideration of our rebuttal, including the additional experimental results, in your final evaluation of our work.
> >
> > Thank you!
> >
> > -Authors

---

> ### Comment · Area_Chair_ufGy · 2024-06-03
> **Please respond**
>
> Hello reviewer,
>
> As a reminder, the discussion period ends on Thursday, June 6. Please take a look at the author's rebuttal and acknowledge if they have adequately addressed your concerns about the paper.
>
> Thanks,
>
> AC

---

### Official Review · Reviewer_cZpL · 2024-05-11

**Rating:** 6
**Confidence:** 3
**Ethics Flag:** 1

**Summary:**

This work introduces erase-and-check to defend against adversarial prompts with certifiable safety guarantees. Given a prompt, this method erases tokens individually and inspects the resulting subsequences using a safety filter, declaring it harmful if any of the subsequences are detected as harmful. They use Llama2 and DistilBERT for safety filters and design their method to certifiably defend against three attack modes: adversarial suffix, adversarial insertion, and adversarial infusion. Then, they propose three empirical defenses: i) RandEC, a randomized subsampling version of erase-and-check; ii) GreedyEC, which greedily erases tokens that maximize the softmax score of the harmful class; and iii) GradEC, which uses gradient information to optimize the tokens to erase.

**Reasons To Accept:**

1. The method is simple and straightforward.
2. The experiments are quite comprehensive, supporting the arguments.
3. The paper is clear and well-structured.

**Reasons To Reject:**

Limitation: Since this method needs to erase tokens from the end and checks the resulting subsequences using a safety filter, thus I think it has a quite high query complexity. In addition, to implement this method effectively, users need to know how attackers manipulate the input prompts, like by appending a suffix, which is challenging and not practical.

---

> ### Author Rebuttal · Authors · 2024-05-31
>
> 1. "... it has a quite high query complexity."
>
>    While our certified defense indeed has a high query complexity, this is not the case with the three empirical variants of our erase-and-check procedure, namely, RandEC, GreedyEC, and GradEC. These approaches enable a trade-off between detection performance and running time. For instance, RandEC achieves detection accuracy ($> 90\%$) close to the certified accuracy while significantly reducing the running time (see Fig. 4). Certified defenses in other domains, such as computer vision, also exhibit high query complexity, underscoring the computational challenges inherent to this problem. For example, randomized smoothing, a prominent certified robustness method for image classifiers, aggregates predictions over a large number of samples to obtain certified guarantees [1].
>
>     Certifying a prompt against all possible adversarial token insertions without imposing any restrictive assumptions on the safety filter is a very difficult problem. Although certain assumptions about the safety filter may help reduce the query complexity, such assumptions may not always be applicable in practice and could substantially limit the scope of the certified defense.
>
>     [1] Certified Adversarial Robustness via Randomized Smoothing, Cohen et al, 2019.
>
> 2. "...users need to know how attackers manipulate the input prompts..."
>
>    Our framework is designed to generate certificates even when the location of adversarial tokens is unknown, a scenario we address with the infusion mode. This mode encompasses the other two modes—suffix and insertion—as specific instances. We study the special cases of suffix and insertion modes separately to demonstrate that if we have some knowledge of how the adversarial tokens are inserted (e.g., as suffix, prefix, etc.), we could use this information to improve the efficiency of our method. Furthermore, we note (on page 5) that existing adversarial attacks, such as GCG and AutoDAN, fall within the suffix and insertion attack modes. To our knowledge, there are no existing attacks that operate within the infusion mode. We study the infusion mode to showcase the versatility of our framework and demonstrate that it can tackle new threat models that emerge in the future.

---

> ### Comment · Area_Chair_ufGy · 2024-06-03
> **Please respond**
>
> Hello reviewer,
>
> As a reminder, the discussion period ends on Thursday, June 6. Please take a look at the author's rebuttal and acknowledge if they have adequately addressed your concerns about the paper.
>
> Thanks,
>
> AC

---

### Official Review · Reviewer_fjZa · 2024-05-19

**Rating:** 5
**Confidence:** 4
**Ethics Flag:** 1

**Summary:**

This paper addresses the vulnerability of large language models (LLMs) to adversarial attacks by introducing "erase-and-check," a framework that provides certifiable safety guarantees. The method involves individually erasing tokens from a prompt and using a safety filter to inspect the subsequences, marking the prompt as harmful if any subsequence is detected as harmful. The authors also propose three efficient defenses, "RandEC, GreedyEC, and GradEC," inspired by erase-and-check. Extensive evaluations demonstrate the effectiveness of these methods against state-of-the-art adversarial attacks.

**Questions To Authors:**

See above weaknesses.

**Reasons To Accept:**

- This paper presents the first framework to defend against adversarial prompts with certifiable safety guarantees.
- The paper is well-written and easy to understand.
- The erase-and-check strategy is simple and effective.
- The authors provide comprehensive experiments across all three threat models, demonstrating their method's effectiveness.

**Reasons To Reject:**

- The method assumes that erased subsequences of safe prompts are always safe, but there are counterexamples (e.g., "Do not do '[SomethingEvil]'" vs. "do '[SomethingEvil]'").
- The proposed method has high query complexity.
- The study should explore more models, such as GPT-3.5, GPT-4, and Llama-2-Chat-7B/13B/70B.

---

> ### Author Rebuttal · Authors · 2024-05-31
>
> 1. While safe prompts that contain harmful subsequences are theoretically possible, they are rare in practice. The high empirical accuracy of erase-and-check on safe prompts (Figure 3) shows that this assumption is valid in the overwhelming majority of cases. We also observe similar performance on safe prompts for other LLMs, such as Llama-3 and Llama-2 13B, presented later in this rebuttal.
>
> 2. While the query complexity is high for some of the certified defenses, this is not the case with the empirical variants: RandEC, GreedyEC, and GradEC. These approaches enable a trade-off between detection performance and running time. For instance, RandEC achieves detection accuracy ($> 90\%$) close to the certified accuracy while significantly reducing the running time (see Fig. 4). Certified defenses in other domains, such as computer vision, also exhibit high query complexity, underscoring the computational challenges inherent to this problem. For example, randomized smoothing, a prominent certified robustness method for image classifiers, aggregates predictions over several samples to obtain certified guarantees [1].
>
>     Certifying a prompt against all possible adversarial token insertions without imposing any restrictive assumptions on the safety filter is a very difficult problem. Although certain assumptions about the safety filter may help reduce the query complexity, such assumptions may not always be applicable in practice and could substantially limit the scope of the certified defense.
>
>     [1] Certified Adversarial Robustness via Randomized Smoothing, Cohen et al, 2019.
>
> 3. In the following, we evaluate the performance of erase-and-check with GPT-3.5, Llama-2 13B, and Llama-3 8B (the latest Llama model) as the safety filter. We report the certified accuracy on the harmful prompts and the empirical accuracy on the safe prompts in the suffix mode. We also include the corresponding results for the Llama-2 7B and DistilBERT models from our paper for comparison. Larger models were excluded due to their high computational and API costs.
>
> | LLM | Cert Acc |
> |------|------------|
> | GPT-3.5 | 100 |
> | Llama-3 8B | 98 |
> | Llama-2 13B | 99 |
> | Llama-2 7B | 92 |
> | DistilBERT | 99 |
>
> Empirical accuracy:
>
> | Max Erase Length | 0 | 10 | 20 | 30 |
> |----------|--|--|--|--|
> | GPT-3.5 | 99 | 87 | 86 | 91 |
> | Llama-3 8B | 100 | 99 | 98 | 98 |
> | Llama-2 13B | 99 | 96 | 96 | 96 |
> | Llama-2 7B | 100 | 98 | 97 | 97 |
> | DistilBERT | 100 | 100 | 100 | 100 |

---

> > ### Comment · Reviewer_fjZa · 2024-06-05
> >
> > Have the authors considered scenarios in which the prompt generated by the model contains nonsensical symbols? How can we certify the quality of such prompts?

---

> > > ### Author Response · Authors · 2024-06-05
> > >
> > > We are unsure what a “prompt generated by the model” means in this context. We would like to clarify that the harmful prompts certified by our erase-and-check procedure are not supposed to be generated by a language model but are assumed to be written by a human. Also, the models employed as safety filters in our procedure do not generate any prompts. They are only used to classify an input prompt as harmful or not harmful.
> > >
> > > In the setting of automated adversarial prompt generation attacks like GCG [1] and AutoDAN [2], the harmful prompt is typically assumed to be written by a human. For instance, a harmful prompt written by a human might state, “Develop a strategy for hacking into a government database and stealing sensitive information.” Given such a prompt, an attack like GCG would produce an adversarial sequence, such as “hilt thou ordinary the *!123,” which, when appended to the harmful prompt, enables it to evade a model’s safety mechanisms and avoid being detected as harmful.
> > >
> > > Given a harmful prompt P, our erase-and-check procedure can certify against any adversarial prompt P + alpha up to a certain length of alpha (referred to as the certified length/size), regardless of the tokens present in alpha. This means that our method can also certify against adversarial sequences alpha that contain nonsensical symbols.
> > >
> > > [1] Universal and transferable adversarial attacks on aligned language models, Zou et al., 2023
> > > [2] AutoDAN: Generating stealthy jailbreak prompts on aligned large language models, Liu et al., 2023.

---

> ### Comment · Area_Chair_ufGy · 2024-06-03
> **Please respond**
>
> Hello reviewer,
>
> As a reminder, the discussion period ends on Thursday, June 6. Please take a look at the author's rebuttal and acknowledge if they have adequately addressed your concerns and questions about the paper.
>
> Thanks,
>
> AC

---

### Author Response · Authors · 2024-05-31
**Authors' comment**

Dear reviewers,

Thank you for dedicating your time and effort to reviewing our work. Given the character limit in the rebuttals, we could focus only on addressing the concerns raised in the review. We are writing to express our gratitude for your commitment to helping us improve our work. Your insightful comments and suggestions have been crucial in enhancing the quality of our paper.

We hope we have adequately addressed your concerns in our rebuttal. We will include all new experimental results and analyses in the revised version of our paper. If you have any further questions or comments, please do not hesitate to let us know.

Thank you!

-Authors

---

### Decision · Program_Chairs · 2024-07-10

**Decision:**

Accept

**Comment:**

The paper presents a method for certifiable safety against adversarial prompts. The reviewers note the novelty of the approach, which is the first to deeply explore defenses against adversarial prompts. They also note that the paper is clearly written with comprehensive experiments. The main concerns are that only a couple models are explored, there is insufficient analysis of failure cases, and that the method is limited because it only works on adversarial suffixes. However, given the overall novelty of the problem, I recommend the paper be accepted.